# A Realistic Simulation Framework for Learning with Label Noise

Keren Gu*  Xander Masotto*  Vandana Bachani*

Balaji Lakshminarayanan†  Jack Nikodem*  Dong Yin*

* DeepMind,  † Google Research, Brain Team

{kerengu, xanderm, vbachani, balajiln, nikodem, dongyin}@google.com

## Abstract

We propose a simulation framework for generating realistic instance-dependent noisy labels via a pseudo-labeling paradigm. We show that this framework generates synthetic noisy labels that exhibit important characteristics of the label noise in practical settings via comparison with the CIFAR10-H dataset. Equipped with controllable label noise, we study the negative impact of noisy labels across a few realistic settings to understand when label noise is more problematic. Additionally, with the availability of annotator information from our simulation framework, we propose a new technique, Label Quality Model (LQM), that leverages annotator features to predict and correct against noisy labels. We show that by adding LQM as a label correction step before applying existing noisy label techniques, we can further improve the models' performance.

## 1 Introduction

In many applications, training machine learning models requires labeled data. In practice, the training data labeled by human raters are often noisy, leading to inferior model performance. The study of learning in the presence of label noise dates back to the eighties [Angluin and Laird, 1988], and still receives significant attention in recent years [Natarajan et al., 2013, Reed et al., 2014, Malach and Shalev-Shwartz, 2017, Han et al., 2018, Li et al., 2020a].

In the research community, some datasets with real noisy human ratings are available, such as Clothing 1M [Xiao et al., 2015], Food 101-N [Lee et al., 2018] (only a small subset has clean labels), WebVision [Li et al., 2017a], and CivilComments [Borkan et al., 2019], which allow testing approaches that address label noise. However, since the level and type of label noise in these datesets cannot be controlled, it becomes hard to conduct ablation study to understand the impact of noisy labels. As a result, the majority of research work in this area uses benchmark datasets generated by simulations. For example, many prior works simulate noisy labels by flipping the labels according to certain transition matrix [Natarajan et al., 2013, Khetan et al., 2017, Patrini et al., 2017, Han et al., 2018, Hendrycks et al., 2018], independently from the model inputs, e.g., the raw images. However, this type of random label noise may not be an ideal way to simulate realistic noisy labels, since the errors in human ratings are often instance-dependent, i.e., harder examples are easier to get wrong labels, whereas the noisy labels generated by random flipping do not have this type of dependency, even if the transition matrix is asymmetric, i.e., class-conditional. In addition, in many applications, we often have additional features of the raters, such as tenure, historical biases, and expertise level [Cabitza et al., 2020]. Leveraging these features properly can potentially lead to better

model performance. However, neither the commonly used public datasets with human ratings nor the synthetic datasets created by random label noise have such rater features available.

In this work, we focus on creating realistic benchmarks for the research on label noise. We propose a simulation method that is easy to implement, more realistic than random label flipping, and can convert any commonly used public dataset with clean labels into a noisy label dataset with additional rater features. More specifically, we propose a simulation method based on a pseudo-labeling paradigm: given a dataset with clean labels, we use a subset of it to train a set of models (rater models), and use them to label the rest of the data. In this way, we obtain a dataset whose size is smaller than the original one with clean labels, but with multiple instance-dependent noisy labels. Moreover, some characteristics of the rater models, such as the number of training epochs, the number of samples used, the validation performance metrics, and the number of parameters in the model can be used as a proxy for the rater features.

We note that this simulation approach is very similar to self-training in semi-supervised learning [Chapelle et al., 2006]. In the research on label noise, methods inspired by semi-supervised learning have been adopted in several prior works for training robust models [Han et al., 2018, Li et al., 2020a] or generate synthetic noisy label dataset [Lee et al., 2019, Robinson et al., 2020]. We intend to exploit this approach for both providing a comprehensive study of how practical label noise affects the performance of machine learning models, and the research of better training algorithm in the presence of label noise. Our main contributions are summarized as follows:

- We propose a pseudo-labeling simulation framework for learning with realistic label noise. We provide detailed description, including the generation of rater features (Section 2.2).

- We propose a systematic protocol for evaluating the synthetic dataset generated by our framework. The evaluation protocol focuses on testing whether the synthetic datasets exhibit some important characteristics of realistic label noise (Section 2.3).

- We study the negative impact of label noise on deep learning models using our synthetic datasets. We find that noisy labels are more detrimental under class imbalanced settings, when pretraining is not used, and on tasks that are easier to learn with clean labels (Section 3).

- We propose a label correction approach, named Label Quality Model (LQM), that leverages rater features to significantly improve model performance. We also show that LQM can be combined with other existing noisy label techniques to further improve the performance (Section 4).

Moreover, we examine the performance of several existing noisy label algorithm on our synthetic datasets without LQM label correction. We find that the behavior of these techniques on our synthetic datasets is different from the datasets generated by independent random label flipping. We present these results in Appendix A.

## 2 Generating synthetic datasets with realistic label noise

In this section, we discuss the formulation of generating synthetic noisy labels, and provide details for the dataset generation procedure and the methods we use to evaluate whether the synthetic datasets share certain characteristics of real human labeled data.

### 2.1 Formulation

We consider a $K$-class classification problem with input space $\mathcal{X}$ and label space $\mathcal{Y} = \{1, \ldots, K\}$. In addition, we assume that there is a rater space $\mathcal{R}$, with each element being the feature of a rater who can label any element in $\mathcal{X}$. Suppose that there is an unknown distribution over $\mathcal{X} \times \mathcal{Y} \times \mathcal{R} \times \mathcal{Y}$m, and each tuple $(x, y^*, r, y)$ in this space corresponds to the input feature of an example $x$, clean label of the example $y^*$, a rater $r$, and the label $y$ provided by the rater.

The problem of generating synthetic noisy labels can be modeled as generating a noisy label $y$ given a pair of input feature and clean label $y^*$. Ideally, the probability distribution of the noisy label $y$ should depend on all of $x$, $y^*$, and $r$, i.e., we should generate $y$ according to $p(y \mid x, y^*, r)$. This means that the label noise should depend on the input—harder and more nuanced examples such as blurred images are more likely to have incorrect labels, as well as the rater—raters with higher expertise level are less likely to make mistakes.

However, many prior studies on generating synthetic noisy labels ignore such dependency on $x$ and $r$ and only generate $y$ according to $y^*$. Here, we specify three approaches for generating noisy labels.

- *Independent random flipping.* In this method, with probability $\delta$, the label of each example is flipped to an incorrect one, uniformly chosen from all the other $K-1$ labels [Zhang et al., 2021, Rolnick et al., 2017, Han et al., 2018]. The method is sometimes called symmetric label noise. More specifically, we have $p(y = k \mid y^*) = (1-\delta)\mathbf{1}(k = y^*) + \frac{\delta}{K-1}\mathbf{1}(k \neq y^*)$.

- *Class-conditional random flipping.* In this method, we assume that there is a stochastic matrix $T \in \mathbb{R}^{K \times K}$. The $i$-th row of $T$ corresponds to the probability distribution of the noisy label $y$ given that the clean label $y^* = i$, i.e., $p(y = j \mid y^* = i) = T_{i,j}$. This method is sometimes called the asymmetric label noise, and is usually considered more realistic than symmetric noise, since classes that are semantically close are more likely to be confused than classes that have clearer decision boundaries. As mentioned in Section 1, this method has been used in many prior works [Angluin and Laird, 1988, Han et al., 2018, Zhang et al., 2017, Wang et al., 2019, Jiang et al., 2018]; the matrix can be designed with human knowledge or estimated from a small subset of clean data [Patrini et al., 2017, Hendrycks et al., 2018]. Here, we emphasize that the noisy labels in class-conditional label flipping still do not depend on the input feature $x$ and the rater $r$.

- *Instance-dependent*, i.e., generating noisy labels according to $p(y \mid x, y^*, r)$. The method that we propose in this paper satisfies this criterion.

## 2.2 Dataset generation

In our framework, we first identify a public dataset that we would like to generate noisy labels for, e.g., CIFAR10 [Krizhevsky and Hinton, 2009] for image classification. We observe that many public datasets already have default training, validation, and test splits. For those without a validation split, we can randomly partition the training data into training and validation splits. We note that in our paper we assume that public datasets have "clean" labels. We acknowledge that many widely used public datasets such as CIFAR10 or ImageNet [Deng et al., 2009] may have mislabeled data points [Northcutt et al., 2021b]; however, the amount of label noise in these public datasets is significantly smaller than what the noisy label research community usually consider [Han et al., 2018, Lee et al., 2019], including our work. Therefore, we believe it is reasonable to consider the labels in public datasets as clean, i.e., less noisy, labels, and we do not expect the label noise in public datasets changes the conclusions in our paper.

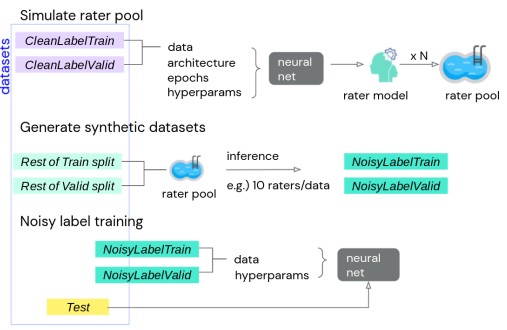

Figure 1: Pseudo-labeling paradigm for simulating realistic noisy labels.

We further split the training and validation splits into two disjoint sets, respectively. More specifically, we partition the training set into *CleanLabelTrain* and *NoisyLabelTrain*, and the validation set into *CleanLabelValid* and *NoisyLabelValid*. We use the data in *CleanLabelTrain* with clean labels to train a set of rater models, which can be any standard models for the problem domain. The data in the *CleanLabelValid* split can be used to evaluate the rater models. For example, the test accuracy with respect to the clean labels on the *CleanLabelValid* split can be used as a feature of a rater model. We can obtain a pool of rater models by choosing different architectures, training epochs, and other training configurations, which can all be used as *rater features*. Then we use all or a subset of models from the rater pool to run inference on the data in the *NoisyLabelTrain* and *NoisyLabelValid* splits. In this way we obtain multiple noisy labels for every data in these two splits, and we replace the clean labels with these noisy labels. We find that in order to control the amount of label noise in these two splits, it is important to train a diverse set of rater models using different combinations of architectures, training steps, learning rate, and batch size. The details for the rater models that we use throughout this paper are provided in Appendix B. To perform label noise research, we can use the *NoisyLabelTrain* split to train models and use the *NoisyLabelValid* split for hyperparameter tuning.[1] For the *Test* split, we use the original clean labels. We illustrate our framework in Figure 1.

---

[1]The *NoisyLabelValid* split also contains noisy labels, which may affect the hyperparameters that we select. Understanding the impact of label noise in the validation set is beyond the scope of this paper and will be a future direction.

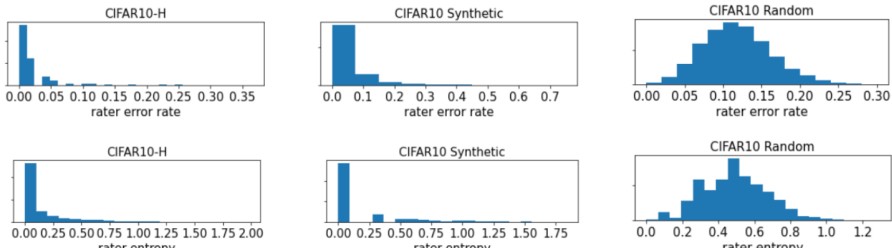

Figure 2: Qualitative comparison among real human labels, our pseudo-labeling framework, and random label flipping. First row: histogram of rater error rate; second row: histogram of rater entropy. Left column: real human labels in CIFAR10-H [Peterson et al., 2019]; middle column: synthetic datasets generated by our pseudo-labeling framework (*NoisyLabelTrain* split); right column: dataset generated by random label flipping. Both CIFAR10-H and our synthetic dataset show right-skewed distribution; whereas the dataset with random label flipping does not.

In most of our experiments, the sizes of the *CleanLabelTrain* and *NoisyLabelTrain* splits are around 50% of the original training and validation splits. However, this ratio can be adjusted depending on the problem of interest. For a synthetic dataset with multiple noisy labels, i.e., the *NoisyLabelTrain* and *NoisyLabelValid* splits, we use the following two metrics to measure the amount of noise in the dataset: (1) overall rater error rate, which is defined as the fraction of the incorrect labels among all the labels given by all the raters, and (2) Krippendorff's alpha (k-alpha) [Hayes and Krippendorff, 2007], which measures the agreement between the raters. We note that the computation of Krippendorff's alpha does not require the clean labels. Usually, datasets with higher k-alpha are less noisy. All model training in this and the following sections are performed on TPUs in our internal cluster.

## 2.3 Dataset evaluation

Once we have the synthetic datasets, the next step is to evaluate how realistic the synthetic datasets that we generate are. Here, we qualitatively show that the distribution of the label noise in our dataset is closer to real human labels, at least when compared to independent label flipping. We evaluate the following two evaluation criteria for noisy label synthetic datasets:

(a) The distribution of rater errors should be qualitatively similar to real human labels.

(b) Label noise should be class-conditional.

**Remark 1** For criterion (a), we note that since the label noise is instance-dependent, when we have multiple labels for every data, the distribution of rater error rate, defined as the ratio of the number of raters that gave wrong labels for a particular data to the total number of raters that labeled this data, should have certain dataset-specific characteristics. For example, in a dataset where the majority of the data are easy to be labeled correctly, while the fraction of the hard examples is relatively small, we should observe a right-skewed distribution, i.e., the majority of the data have small rater error rate, and data with large rater error rates account for the tail of the distribution. The similar trend should also be observed for the entropy (measuring consistency) of the noisy labels. We note that this right-skewed distribution should not be observed in datasets where labels are flipped with a fixed probability, *even if the label flipping is class-conditional*. This is because with random flipping, the rater error rate and rate entropy of the data should concentrate around certain values when we have multiple labels per data.

We emphasize that our evaluation focuses on the overall trend of the distribution of label noise, rather than each individual instance. This means that we do not expect that our synthetic noisy labels closely match a certain human labeled dataset for every single data point. Instead, we qualitatively compare the distribution of the rater error rate (or entropy) and show that the datasets have similar trends. Making the predictions of the rater models (neural networks) more close to human behavior is an interesting problem to study, but is beyond the scope of this paper.

**Remark 2** For criterion (b), we aim to show that the class confusion matrix has non-uniform off-diagonal entries. We acknowledge that this can only show that our method is class-conditional rather than instance-dependent. However, one important difference from prior work is that the confusion among classes in our framework occurs after the rater models are trained; whereas in prior works it needs to be designed by human or estimated from a subset of data.

**Evaluation results** We now proceed to present our evaluation results. We create a synthetic dataset based on the CIFAR10/100 dataset. For **criterion (a)**, in order to obtain real human labels, we use the CIFAR10-H dataset, recently published by Peterson et al. [2019]. This dataset contains the 10K data points from the CIFAR10 test split, and a total of 500K human labels were crowdsourced, i.e., on average around 50 labels for each data. In our proposed simulation framework, we train 10 rater models using the *Clean-*

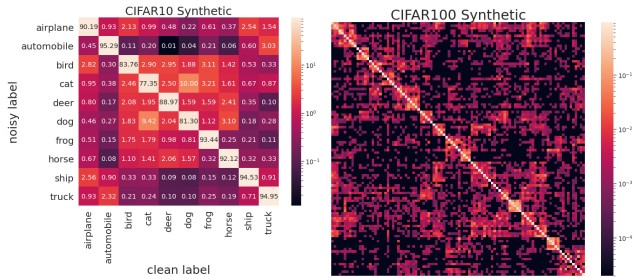

Figure 3: Class confusion matrix for our synthetic dataset. Columns and rows in the CIFAR100 figure (right) are grouped by the 20 coarse labels.

*LabelTrain* split and use them to label the *NoisyLabelTrain* split. Details of our synthetic dataset can be found in Appendix B. For independent random label flipping, we generate 50 labels for each data, and flipping probability $\delta = 10.8\%$. matching our synthetic dataset. As we can see in Figure 2, the synthetic datasets generated by our pseudo-labeling framework produce similar right-skewed distribution to the datasets with real human labels in CIFAR10-H, whereas the datasets generated by random label flipping have different trends. Thus, we conclude that our proposed simulation framework is more realistic on the first evaluation criterion. As for **criterion (b)**, we compute the class confusion matrix two synthetic datasets generated by our method for CIFAR10 and CIFAR100, respectively. As shown in Figure 3, the label noise in our dataset is class-conditional. We note that one can verify that the class confusion matrix of CIFAR10-H is also class-conditional; however, ensuring that the confusion matrix of our synthetic dataset closely matches certain human labeled dataset is non-trivial and is beyond the scope of this paper.

## 3 Impact of realistic label noise on deep learning models

With the realistic synthetic datasets with noisy labels, our next step is to study the impact of noisy labels on deep learning models. Interestingly, there exist different views for the impact of noisy labels to deep neural networks. While most of the recent research works on noisy labels try to design algorithms that can tackle the negative impact of label noise, some other works claim that deep learning models are robust to independent random label noise [Rolnick et al., 2017, Li et al., 2020b] without using sophisticated algorithms. A prominent example is the weak supervision paradigm [Ratner et al., 2016, 2017], where massive training datasets are generated by weak raters and labeling functions. Other lines research indicate that large neural network can easily fit all the noisy labels in the training data [Zhang et al., 2021], while smaller models may be more robust against label noise due to the regularization effect [Advani et al., 2020, Belkin et al., 2019, Northcutt et al., 2021b].

We hypothesize that the negative impact of noisy labels is problem-dependent. While in most cases the incorrect labels can impair models' performance, the impact may depend on factors related to the data distribution and the model. In this section, we choose the following factors to measure the impact of label noise: the class imbalance, the inductive bias of the model (in particular, pretraining vs random initialization), and the difficulty of the task (test accuracy that models can achieve when clean labels are accessible). Note that for better understanding, we decouple these factors with algorithm design: In this section, we choose simple SGD-style training algorithms with cross-entropy loss and focus on analyzing the impact of label noise; the discussion on more sophisticated algorithms to tackle label noise is presented in Sections A and 4. We do not aim to study label aggregation methods either. Instead, in this and the following sections, given a synthetic dataset with multiple noisy labels, we generate a dataset with a *single noisy label* by independently and uniformly selecting a random noisy label for every data point. This is a simulation of the realistic setting where we have a pool of raters and for each data, we choose a random rater from the pool and request a label.

### 3.1 Label noise has higher impact on more imbalanced datasets

One of the important characteristics of many real world datasets is that the classes are usually imbalanced. When the classes are more imbalanced, the impact of noisy labels may become more pronounced since the number of data in the minority classes is already small, and noisy labels can further corrupt these data, making the learning procedure more difficult. We validate this hypothesis in this section. We use two binary classification tasks, PatchCamelyon (PCam) [Veeling et al., 2018,

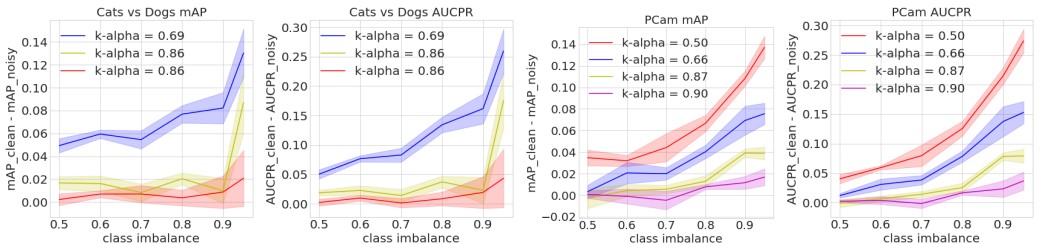

Figure 4: Impact of label noise for tasks with different class imbalance. The x-axis represents class imbalance, measured by the fraction of the majority class. For PCam, we use MobileNet-v1 [Howard et al., 2017] model, and for Cats vs Dogs, we use ResNet50 [He et al., 2016]. The k-alphas correspond to the synthetic datasets before subsampling.

[Bejnordi et al., 2017] and Cats vs Dogs (CvD) [Elson et al., 2007]. We generate synthetic noisy label datasets with different k-alphas, and for each of these datasets, we subsample the two classes to create several smaller datasets with different class imbalance but the same total number of data. We note that here we control the class imbalance to be the same for all of the *NoisyLabelTrain*, *NoisyLabelValid*, and *Test* splits. We train models with clean and noisy labels and use the difference in mean average precision (mAP) [Zhang and Zhang, 2009] and area under the precision-recall curve (AUCPR) [Raghavan et al., 1989] as the indicators for the impact of label noise. The results are shown in Figure 4. As we can see, the impact of label noise becomes more significant as the classes become more imbalanced.

## 3.2 Pretraining improves robustness to label noise

One model training technique that is often used in practice, especially for computer vision and natural language tasks, is to pretrain the models on some large benchmark datasets and then fine-tune them using the data for specific tasks. It has been observed that model pretraining can improve robustness to independent random label noise [Hendrycks et al., 2019] and the web label noise considered by Jiang et al. [2020]. Here we show that this can still be observed in our synthetic framework. A simple explanation is that model pretraining adds strong inductive bias to the models and thus they are less sensitive to a fraction of noisy labels during fine-tuning.

We validate this hypothesis using two datasets, Cats vs Dogs (CvD) and CIFAR10. For both datasets, we generate three synthetic noisy label datasets using our framework with different rater error rates. We compare the test accuracy on the *Test* split (with clean labels) between the models that are trained from random initialization and those that are fine-tuned from models pretrained on ImageNet [Deng et al., 2009]. We experiment with three different architectures, including Inception-v4 [Szegedy et al., 2017], ResNet152, and ResNet50 [He et al., 2016]. As we can see in Figure 5, models that are pretrained on ImageNet achieve better test accuracy. In addition, for pretrained models, the test accuracy tends to drop more slowly compared to models that are trained from random initialization as we increase the amount of noise (rater error rate).

Meanwhile, we also observe that ImageNet pretraining does not improve the test accuracy under noisy labels for the PatchCamelyon dataset. This can be explained by the fact that the PatchCamelyon dataset consists of histopathologic scans of lymph node sections, and these medical images have very different distribution from the data in ImageNet. Therefore, the inductive bias that the model learned from ImageNet pretraining may not be helpful on PCam.

## 3.3 Easier tasks are more sensitive to label noise

We also study the impact of label noise on tasks with different difficulty levels (the test accuracy models can achieve when clean labels are accessible). Our hypothesis here is that when a task is already hard to learn even given clean labels, then the impact of label noise is smaller. The reason can be that when a classification task is hard, the data distributions of different classes are relatively close such that even if some data are mislabeled, the final performance may not be heavily impacted. On the contrary, label noise may be more detrimental to easier tasks as the data distribution can significantly change when well-separated data points get mislabeled. We validate this hypothesis with two experiments.

**Setup.** Our first experiment involves two binary classification tasks, i.e., PatchCamelyon (PCam) with MobileNet-v1 [Howard et al., 2017] and Cats vs Dogs with ResNet50 [He et al., 2016]. We

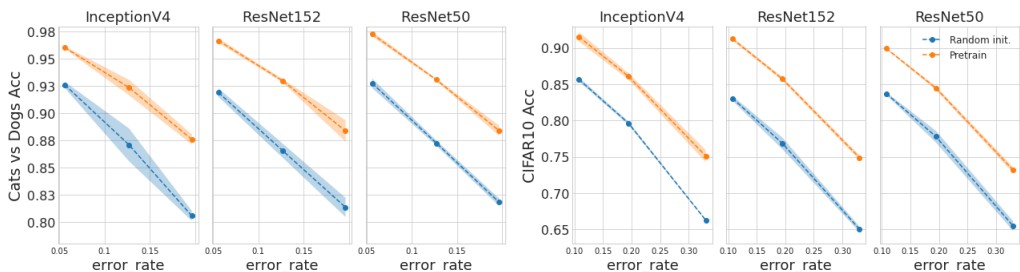

Figure 5: Pretrained models achieve better test accuracy on CvD and CIFAR10. Moreover, as the amount of label noise increases, the amount of test accuracy drop is smaller for pretraining (e.g. the slope is smaller) than training from random initialization.

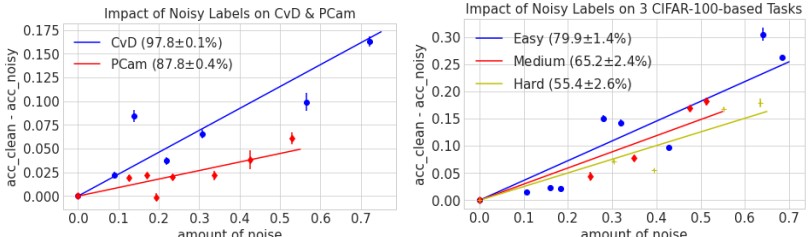

Figure 6: Impact of label noise on tasks with different difficulty levels. The numbers in the legend correspond to test accuracies when training with clean labels for every task. The x-axis represents the amount of noise, measured by $1.0-$ k-alpha. The y-axis represents the negative impact of label noise, measured by the difference in test accuracy when training with clean and noisy labels. Scattered points represent the pairs of noise level and the accuracy drop, and solid lines show the linear fit of the scattered points in the same color.

generate synthetic noisy label datasets with different k-alphas using our framework, and compare the accuracies when the models are trained with clean and noisy labels. We observe that CvD is easier than PCam (clean label accuracy $97.8 \pm 0.1\%$ vs $87.7 \pm 0.4\%$). In our second experiment, we design three 20-way classification tasks with the same number of data but different difficulty levels by subsampling different classes from the CIFAR100 dataset. We call the three tasks the *easy*, *medium*, and *hard* tasks. Details for the design of the three tasks are provided in Appendix C, and we observe that with clean labels, we can obtain test accuracies of $79.9 \pm 1.4\%$, $65.2 \pm 2.4\%$, and $55.4 \pm 2.6\%$ for the three tasks, respectively. We generate synthetic datasets with different amounts of noise, measured by k-alpha and use the MobileNet-v2 model [Sandler et al., 2018].

**Results.** We study the impact of label by measuring the absolute difference in test accuracy when training with clean and noisy labels. The results are shown in Figure 6. As we can see, the impact of noisy labels is higher on the easy task: On CvD, the drop in test accuracy grows faster as we increase the amount of label noise (indicated by $1.0-$ k-alpha) compared to PCam, and similar phenomenon can be observed on the three CIFAR100-based tasks.

## 4 Leveraging rater features: Label Quality Model

Existing work in the noisy label literature commonly assumes that training labels are the only output of the data curation process. In practice however, the data curation process often produces a myriad of additional features that can be leveraged in downstream training, e.g., which rater is responsible for a given label, as well as that rater's tenure, historical errors, and time spent on a given task. With our proposed method of simulating instance-dependent noisy labels via rater models, we can additionally simulate these rater features by extracting metadata from the rater models, e.g., the number of epochs used to train the rater models is a proxy for rater tenure. Another common practice in label curation is assigning multiple raters for a single example. This is commonly used to reduce the label noise via aggregation, or to evaluate the performance of individual raters against the pool. This practice assumes that agreement among multiple raters are more accurate than individual responses.

With understanding of practical data collection setup, we introduce a technique for training with noisy labels, which we coin *Label Quality Model* (LQM). LQM is an intermediate supervised task aimed at predicting the clean labels from noisy labels by leveraging rater features and a paired subset for supervision. The LQM technique assumes the existence of rater features and a subset of training data with both noisy and clean labels, which we call *paired-subset*. We expect that in real world scenarios some level of label noise may be unavoidable. LQM approach still works as long as the clean(er) label is less noisy than a label from a rater that is randomly selected from the pool, e.g., clean labels can be from either expert raters or aggregation of multiple raters. LQM is trained on the paired-subset using rater features and noisy label as input, and inferred on the entire training corpus. The output of LQM is used during model training as a more accurate alternative to the noisy labels.

The intuition for LQM is to correct the labels in a rater-dependent manner. This means that by learning the patterns from the paired-subset, we can conduct rater-dependent label correction. For example, LQM can potentially learn that raters with a certain feature often mislabel two breeds of dogs, then it can possibly correct these two labels from similar raters for the rest of the data. Below we formally present the details of LQM.

**Algorithm design.** Formally, let $D := \{(x_i, y_i, r_i)\}_{i=1}^N$ be a noisy label dataset, e.g., the *Noisy-LabelTrain* split,[2] where $x_i$ is the input, $y_i$ is the one-hot encoded noisy label, and $r_i$ is the rater feature corresponding to $y_i$. Let $D_{ps} = \{(x_j, y_j, r_j, y_j^*)\}_{j=1}^M$ be the paired-subset, and $y_j^*$ be a more accurate label than $y_j$. We usually have $M \ll N$. We propose to optimize a parameterized model $\mathsf{LQM}(\theta; x, r, y)$ to approximate the conditional probability $P(y^*|x, r, y)$ using $D_{ps}$. We note that LQM leverages all the information from the input $x$, rater features $r$, and the noisy label $y$.

Once we have the LQM, we proceed to tackle the main task using the noisy label dataset $D$. Instead of trying to predict $P(y_i|x_i)$, we replace the noisy labels $y_i$ with the outputs of LQM and train a model to predict $P(\mathsf{LQM}(\theta; x_i, r_i, y_i)|x_i)$. From experimentation, we find that by interpolating between noisy label $y_i$ and the output of LQM produces even stronger results. Therefore, we recommend training with target $\tilde{y}_i = \gamma \mathsf{LQM}(\theta; x_i, r_i, y_i) + (1 - \gamma)y_i$, where $\gamma$ is a hyperparameter between $0$ and $1$ and can be selected using the validation set. This is particularly helpful for datasets with a large number of classes such as CIFAR100, since it prevents the training target from getting too far from the original labels $y_i$. Moreover, since $\tilde{y}_i$ specifies a distribution over the labels, we can also sample a single one-hot label according to the distribution $\tilde{y}_i$ as the target.

We use a small set of rater features in the simulated framework, such as the accuracy of the rater model on *CleanLabelValid*, the number of epochs trained, and the type of architecture. In addition, we also use the paired-subset to empirically calculate the confusion matrix for each rater and use it as a feature for the rater. Instead of training LQM with raw input $x$, we first train an auxiliary image classifier $f(x)$ and train LQM using the output logits of $f(x)$. The auxiliary classifier can be trained over either the full noisy dataset $D$ or the paired-subset $D_{ps}$. We find that the better option depends on the task and the amount of noise present. In our experimentation, we train $f(x)$ on both dataset options and select the better one. Given that LQM has fewer training examples, using an auxiliary image classifier significantly simplifies training.

**Experiment setup and results.** For uniformity, we assume $M = 0.1N$, i.e. 10% of training data has access to a clean label in all of our following experiments. For the main prediction model, i.e., $P(\mathsf{LQM}(\theta; x_i, r_i, y_i)|x_i)$, we use the ResNet50 architecture. For the auxiliary model $f(x)$, we use MobileNet-v2. The LQM itself is trained using a one-hidden-layer MLP architecture with cross-entropy loss. The number of hidden units in the MLP is chosen in $\{8, 16, 32\}$ as a hyperparameter. We conduct the following two experiments (see the exact numbers for the results in Appendix D).

- **LQM vs baseline.** First, we compare the performance of models trained with LQM and the baseline models that are trained using vanilla cross-entropy loss without leveraging rater features. Since LQM has access to clean labels of 10% of the data, for fair comparison, we ensure that the baseline models also have access to the same number of clean labels. The comparison is presented in Figure 7. As we can see, with rater features and the label correction step, in many cases, especially in the medium and high noise settings, LQM outperforms the baseline.

- **Combining LQM with other techniques.** In the second experiment, we investigate the conjunction of LQM with other noisy labels techniques. We hypothesize that, depending on the

---

[2]As mentioned in Section 2.2, the size of the *NoisyLabelTrain* split is around 50% of the training split of the original dataset.

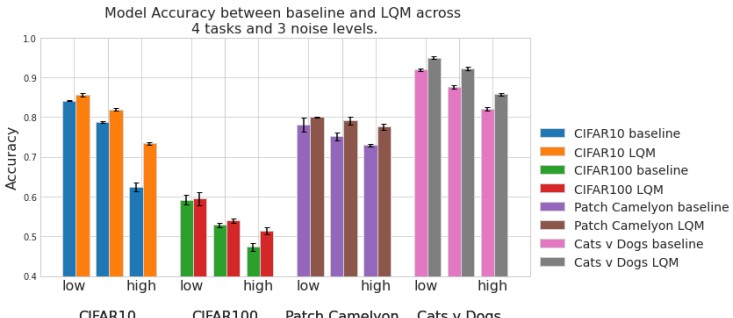

Figure 7: Training with LQM adjusted labels outperforms baselines across all datasets. The improvement from LQM is more prominent in medium and high noise settings. LQM models are trained by randomly sampling a one-hot label from LQM output distribution, and tuning the $\gamma$ parameter.

technique, LQM may be correcting a different kind of noise from existing techniques, and thus can potentially lead to further performance improvement. To combine LQM with another technique, we sample a hard label from the soft distribution specified by $\tilde{y}_i$, and apply other noisy labels techniques on top of the sampled hard label. We consider the following 4 techniques: Bootstrap [Reed et al., 2014], Co-Teaching [Han et al., 2018], cross-entropy loss with Monte Carlo sampling (MCSoftMax) [Collier et al., 2020], and MentorMix [Jiang et al., 2020]. We find that on CIFAR10 and CIFAR100, combining LQM with other techniques usually lead to further performance improvement. The improvement can also be observed in the high noise setting for CvD. The results are illustrated in Figure 8.

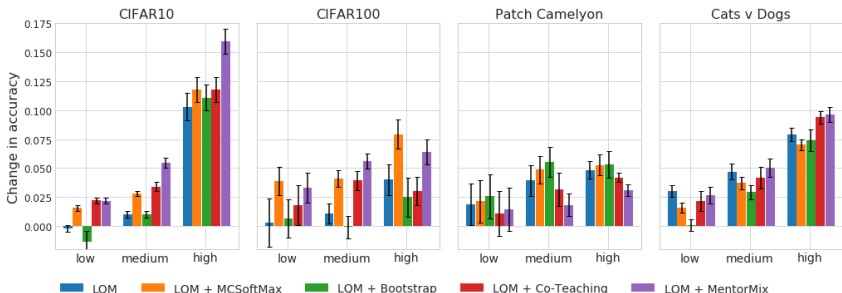

Figure 8: Accuracy improvement of LQM and the combinations of LQM and other techniques compared to the baseline. In many settings (CIFAR10, CIFAR100 and high noise setting in CvD), combining other techniques with LQM further improves the test accuracy. In other cases (PCam and low/medium noise for CvD), the performance gain is less significant.

As a final note, since LQM assumes access to a subset of data with clean labels, and also uses an auxiliary classifier $f(x)$, it has some similarity with semi-supervised learning (SSL). We notice that several state-of-the-art SSL techniques such as FixMatch [Sohn et al., 2020], UDA [Xie et al. [2019], self-training with noisy student [Xie et al., 2020] use specifically designed data augmentations that are only suitable for specific types of data, whereas LQM can be applied to any type of data as long as we have rater features. We also expect that combining certain SSL techniques (e.g., data augmentation and consistency training) can further improve the results; however, these extensions are beyond the scope of this paper.

## 5 Additional related work

There is a large body of literature on learning with noisy labels. We mentioned several related work in the previous sections and it is certainly not an exhaustive list. Since our focus is a more realistic simulation framework for noisy label research, we first review prior works that try to simulate noisy labels using methods beyond random label flipping or permutation. As mentioned in Section 1, we are aware that two prior works by Lee et al. [2019] and Robinson et al. [2020] that also use similar pseudo-labeling paradigm to generate synthetic datasets with noisy labels. Seo et al. [2019] use a similar idea of nearest neighbor search in the feature space of a pretrained model with clean labels

to generate noisy labels. Compared with these works, our study is much more comprehensive with a diverse set of tasks and model architectures. We conduct a series of analysis on the impact of noisy labels, and propose a method to generate synthetic rater features and use them for improving robustness. These points were not considered in the two prior works. Other approaches to simulating realistic label noise have also been studied in the literature. Jiang et al. [2020] proposes a framework to generate controlled *web label noise*, in which case new images with noisy labels are crawled from the web and then inserted to an existing dataset with clean labels. The framework differs from our approach and the two frameworks should be considered complementary for generating realistic noisy label datasets. In particular, the method by Jiang et al. [2020] is more suitable for web-based data collection, e.g., WebVision [Li et al., 2017a] whereas ours is more suitable for simulating human raters. Moreover, Wang et al. [2018] and Seo et al. [2019] consider open-set noisy labels, where the mislabeled data may not belong to any class of the dataset, similar to Jiang et al. [2020]. Another approach to generating datasets with controllable about of label noise is to first identify a dataset with noisy labels (potentially some public datasets [Northcutt et al., 2021a,b]) and then use the confident learning (CL) method [Northcutt et al., 2021a] to increase or decrease the amount of label noise proportionally to the distribution of real-world label noise in the dataset. The idea is to model the joint distribution of noisy and true labels and then generate the noisy labels based on the noise-increased or noise-decreased joint distribution of noisy and true labels. This differs from our method since we use rater models, which are trained neural networks to generate noisy labels for each instance.

Tackling noisy labels using a small subset of data with clean labels has been considered in a few prior works. Common approaches include pretraining or fine-tuning the network using clean labels [Xiao et al., 2015, Krause et al., 2016], and distillation [Li et al., 2017b]. In loss correction approaches, a subset of clean labels are often used for estimating the confusion matrix [Patrini et al., 2017, Hendrycks et al., 2018]. Veit et al. [2017] propose a method that estimates the residuals between the noisy and clean labels. Ren et al. [2018] use the clean label dataset to learn to reweight the examples. Tsai et al. [2019] combine clean data with self-supervision to learn robust representations. Our approach differs from these prior works since we leverage the additional rater features to learn an auxiliary model that corrects noisy labels in a rater-dependent manner, and can be combined with other techniques to further improve the performance as shown in Section 4.

Learning from multiple annotators has been a longstanding research topic. The seminal work by Dawid and Skene [1979] uses the EM algorithm to estimate rater reliability, and much progress has been made since then [Raykar et al., 2010, Zhang et al., 2014, Lakshminarayanan and Teh, 2013]. Rater features is commonly available in many human annotation process. In crowdsourcing, several prior work focus on estimating the reliability of raters [Raykar et al., 2010, Tarasov et al., 2014, Moayedikia et al., 2019], and rater aggregation [Vargo et al., 2003, Chen et al., 2013]. Item response theory [Embretson and Reise, 2013] from the psychometrics literature uses a latent-trait model to estimate the proficiency of raters and the difficulty of examples, and has a similar underlying principle to our work.

Our method is also broadly related to several other lines of research. Training a pool of rater models is similar to ensemble method [Dietterich, 2000], which is usually used to boost test accuracy [Freund and Schapire, 1997] or improve uncertainty estimation [Lakshminarayanan et al., 2017]. Training new models using noisy labels provided by the rater models is similar to knowledge distillation [Buciluă et al., 2006, Hinton et al., 2015]. Designing instance-dependent noisy label generation framework can be considered as reducing underspecification [D'Amour et al., 2020] in generating label noise.

## 6 Conclusions

We propose framework for simulating realistic label noise based on the pseudo-labeling paradigm. We show that the synthetic datasets that we generated are more realistic than independent random label noise. With our synthetic datasets, we evaluate the negative impact of label noise on deep learning models, and demonstrate scenarios where label noise is more detrimental. Using the rater features from our simulation framework, we propose a new technique, Label Quality Model, that leverages annotator features to predict and correct against noisy labels. Our work demonstrates the importance of using realistic datasets for noisy label research, and we hope our framework can serve as an option for the noisy label research community to develop more efficient methods for practical challenges. One limitation of our framework is that, as discussed in Section 5, it focuses more on simulating human errors, whereas there might be other types of label noise in practical settings that need other simulation methods. Another limitation is that LQM requires the paired-subset containing both clean and noisy labels. This requirement may not be satisfied in some applications.

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
