# OpenReview forum: "A Realistic Simulation Framework for Learning with Label Noise"
_NeurIPS.cc/2021/Track/Datasets_and_Benchmarks/Round1 — Submitted to NeurIPS 2021 Datasets and Benchmarks Track (Round 1)_

### Official Review · Reviewer_mRWv · 2021-07-03
**Interesting idea with well carried and thorough empirical experiments.**

**Rating:** 7
**Confidence:** 4
**Clarity:** The text and figures are clear. The p…

**Strengths:**

1. The proposed simulation framework is technically sound and similar to how human raters annotate real-world data.
2. The experimental design for evaluating the generated dataset is thorough and clean. The experimental results in section 3 reveal the impacts of realistic label noise on different deep learning models.
3. The proposed label quality model (LQM) is demonstrated to largely improve model performance with rater features. I like the idea of using extra information from rater models to enhance the model's performance and this can motivate the future algorithm design on learning with noisy data.

**Weaknesses:**

1. Simulating human errors in data annotation is good, but it would be better to enhance the current framework with other types of errors.
2. The quality of generated pseudo-labels will be highly dependent on how the pool of rater models is constructed. It would be better to dive deeper into what types of models / how many models are needed to create a high-quality rater models' pool.

**Additional Feedback:**

1. One benefit of independent random label flipping is that the simulator can generate pseudo-labels with arbitrary user-defined noisy ratios. It seems to be hard to control the noisy ratio in the proposed simulation framework as the quality of rater models are depending on their architecture designs.
2. Typos: line 322: data are may not / line 177: characteristic -> characteristics / line 64: on the our

**Correctness:**

The proposed approach and constructed synthetic datasets are sound. The empirical experimental analysis is thorough.


**Documentation:**

Yes, the authors have provided sufficient information for reproducibility.

**Ethics:**

No ethics issues.

**Relation To Prior Work:**

A detailed related work section is provided in the paper. The authors clearly distinguish their work from related noisy label simulation frameworks.

**Summary And Contributions:**

In this paper, the task of performing meta-learning based on the unsupervised dataset is considered. The high-level idea is to mimic the way that human raters annotate real-world data via using a diverse pool of rater models to generate pseudo-annotations. The proposed method is straightforward and the synthetic datasets exhibit some important features of realistic label noise. The empirical results reflect the usefulness of the proposed simulation framework. The writing was clear and easy to follow. I like the experimental design and the comparison analysis with independent random label flipping.

---

> ### Author Response · Authors · 2021-07-09
> **Review response**
>
> Thanks for your review and positive feedback.
>
> Regarding other types of label noise, we agree that it would be interesting to study how to design other simulation methods for various types of noise. We mentioned a few related work in Section 5. This will be one of our future directions.
>
> The amount of noise in our framework is controllable. As mentioned in the paper, we need to use diverse combinations of model architectures, training epochs, learning rate, and batch size, etc to create rater models with different quality. The details of these rater models are in Appendix B. In the revised version, we added a sentence "We find that in order to control the amount of label noise in these two splits, it is important to train a diverse set of rater models using different combinations of architectures, training steps, learning rate, and batch size." in Section 2. We agree with the reviewer that controlling the amount of noise in our framework is harder than independent random flipping, but we believe it is worth doing in order to obtain more realistic synthetic data.
>
> We have fixed the typos you mentioned in our local file and will upload the revised version once we address all the comments. We are happy to discuss more during the discussion phase.

---

> > ### Comment · Reviewer_mRWv · 2021-07-14
> > **Thanks for your reply**
> >
> > Thanks for your update and I will keep my score.

---

### Official Review · Reviewer_J3SL · 2021-07-04
**Nice study but contribution remains unclear**

**Rating:** 4
**Confidence:** 4

**Strengths:**

- I appreciate Section 2 which shows to some extent (see below for more comments) that the pseudo-labels from a set of models is in certain characteristic close to that of a set of human raters using the CIFAR10-H dataset. A more closer look would help how close these relations really are

- extensive experiments on the impact of label noise in Section 3 regarding impact of imbalance of classes, pretraining, sensitivity to label noise according to the class difficulty

- the label quality model seems to improve over the standard baseline

**Weaknesses:**

- While I agree  that the aggregation of the results of different models is more correlated with human label noise rather than random label noise, it is not clear on a more individual level if this is true or not e.g. one could have checked the agreement of the entropy of predictions on test points by comparing the overall ranking of the test according to the entropy of predictions. As far as I understand the Peterson et al paper they did this only for a very strong single model on CIFAR10. It remains also unclear if the aggregation of different models is a better predictor for human label noise than the best individual model (the numbers in the confusion table for CIFAR-10 Synthetic are significantly worse than that of a good standard model on CIFAR10. It would be very interesting if there were particular classes or even subsets of classes on which human raters and models agree resp. disagree.
If the authors can provide this stronger evidence then it would be very nice, if not then the authors should avoid suggesting like in Figure 1 that the aggregation of deep models is the same as aggregating human raters.

- the y-axis label in Figure 2 is missing, the x-axis should be the same for all three plots to make the comparison easy. For the y-axis I recommend a logarithmic scale as otherwise the small counts are not visible

- Figure 4 shows curves with different k-alpha but the k-alpha seems to be computed for the synthetic dataset before subsampling, but if the disagreement is depending on the classes then k-alpha after subsampling can be quite different?
Also curves of the same k-alpha or very similar one show quite different behavior already which questions a bit the general statements made

- it seems rather obvious that pre-trained model on a similar task helps and does not help if the task is completely different, the same seems true with the finding that label noise is affecting easier tasks more than already difficult ones

- in Figure 6 (right) the linear fit looks quite noisy

- the result of the LQM with or without improvement seem to be worse than what SSL techniques also relying on pseudo-labeling resp. self-learning like Fixmatch, UDA achieve with only 250 labels on CIFAR10 resp. 2500 labels on CIFAR100 (which seems less than the number of clean labels used in the experiments, even though N seems nowhere specified).
This part requires a much better comparison to also other similar techniques like noisy student SSL
Self-training with Noisy Student improves ImageNet classification, Qizhe Xie, Minh-Thang Luong, Eduard Hovy, Quoc V. Le, CVPR 2020

- the suggestion to use in the end a convex combination of LQM and noisy label is unmotivated and introduces another hyperparameter

- it remains completely unclear what the LQM model really learns, there is also no discussion of this anyhwere




**Additional Feedback:**

If the authors can emphasize more the part in Section 2, then the paper would be better suited for this benchmark/dataset track. The LQM method seems a bit off-topic and there remains the missing comparison to self-training/pseudo-labeling SSL techniques.

**Clarity:**

- I guess that just LQM in Figure 8 means LQM+Bootstrap
- Why is then LQM+Bootstrap improving so strongly for high label noise?

- N is unspecified

- In the appendix it remains unclear what kind of data augmentation is used also the performance of the individual models is not provided - in particular what do these ones achieve just training on the clean data?




**Correctness:**

Apart from the questions and concerns mentiond in other paragraphs, everything seems fine.

**Documentation:**

This seems not to be really a dataset submission but rather a submission on a procedure how to generate more realistic label noise.
In this sense the paper is also a bit off-topic in this track.


**Relation To Prior Work:**

There is a huge literature on this topic and I am not super-familiar with the literature, in particular not the one regarding the aggregation of human labels. Thus it seems ok but it could be that I missed important related work.

**Summary And Contributions:**

The paper proposes to produce label noise by aggregating the pseudo-labels of different models. The main idea is that this form of label noise is a better model for the label noise generated by human raters as it correlates with the difficulty of the instances.
Using this partially controllable form of label noise several hypothesis on the impact of label noise are checked in experiments. Finally, a Label Quality Model is proposed to predict from given noisy labels and "rater information" the true label.

I appreciate the study in Section 2 about the possibility to model human label noise via classifiers the presentation has to be more detailed to really see the correspondence, the findings of Section 3 are known or seem quite obvious, regarding the Label Quality model the final results for CIFAR10 and CIFAR100 are worse than what one can achieve with Semi-supervised Learning Techniques e.g. Fixmatch, UDA etc with only 250 labels.

---

> ### Author Response · Authors · 2021-07-10
> **Review response -- we believe all the issues have been clarified in the response and/or addressed in the revision**
>
> Thanks for your time and comments. We believe all the issues that you raise have been clarified in this response and/or addressed in the revision in our local file. The revised version will be uploaded once we finish addressing all the comments from the reviewers. Here, we provide a point-by-point response to your comments. We are also happy to discuss more during the discussion phase, and we hope you can increase your score after reading our response and revision.
>
> **Weaknesses** section
>
> - *"it is not clear on a more individual level if this is true or not e.g. one could have checked the agreement of the entropy of predictions on test points ... if not then the authors should avoid suggesting like in Figure 1 that the aggregation of deep models is the same as aggregating human raters"* ,  *"the y-axis label in Figure 2 ..."*
>
> We would like to emphasize that the goal of our simulation framework is to generate synthetic noisy labels that are as realistic as possible, rather than making the noisy labels exactly the same as certain datasets collected from human raters. In other words, we show that the **distribution** the label noise in our dataset is closer to real human labels compared to independent random label flipping. Currently, we don’t think ensuring that our rater models (neural networks) make similar prediction errors as human raters for each data point is necessary for our framework, although it could be an interesting future direction. In the revision, in Section 2.3 (the dataset evaluation section, corresponding to Section 2.2 in the original submission), we emphasize that the comparison focuses more on the **trend of the distribution rather than individual instances**, and is more qualitative than quantitative.
>
> We intentionally removed the y-axis in Figure 2 since the histograms have different scales. The important message here is that for our dataset and CIFAR10-H, the shapes of the distribution histograms are both right-skewed, whereas random label flipping has a different shape (concentrated in the middle).
>
> For the confusion matrix, again the main message we have is that both our dataset and CIFAR10-H have non-uniform off-diagonal entries (i.e., class-conditional). We do not aim to ensure that all the entries are close to each other. In fact, the rater error rate in CIFAR10-H (around 5%) is in general smaller than the test error of many typical neural network models; moreover, similar to most prior work on noisy label, the amount of label noise in our datasets is controllable and can be varied by the users. To avoid misunderstanding, in the revision, we removed the confusion matrix of CIFAR10-H and emphasized that in this section we only aim to verify that the label noise in our datasets is class-conditional.
>
> - *"Figure 4 shows curves with different k-alpha but the k-alpha seems to be computed for the synthetic dataset before subsampling ... "*
>
> It is correct to say that downsampling can change the k-alpha. We expect that the dataset with higher k-alpha (more agreement) will also have a higher k-alpha after downsampling. We have 3 (or 4) curves in each subfigure in Fig 4 and each curve has 6 different downsampling ratios, so it would be hard to annotate the k-alpha for each point on these figures.
>
> However, the main message in this figure is that **the impact of label noise increases as class imbalance increases**. So we think the best way to interpret this figure is to look at each curve and observe that as we move from left to right, the curves increase, which validates our claim.
>
> In the first two subfigures, the red and yellow curves correspond to the same original k-alpha (0.86), and are close enough except for the highly imbalanced case (0.95). However, in this case, due to the high class imbalance, the results can have relatively large variance, so we believe the phenomenon in this figure is expected.
>
> - *"it seems rather obvious that pre-trained model on a similar task helps and does not help if the task is completely different ..."*
>
> These observations may be intuitive to some readers. However, several similar studies on different synthetic noisy label datasets have been published (see Hendrycks et al 2019, Jiang et al. 2020 in our reference). We believe it’s worth studying them in our (more realistic) synthetic datasets.
>
> - *"in Figure 6 (right) the linear fit looks quite noisy"*
>
> This is due to the relatively small size of the datasets. We have to sample 20 classes of the CIFAR100 dataset to create these tasks. The left figure (CvD and PCam) uses more data which clearly shows the trend.

---

> > ### Author Response · Authors · 2021-07-10
> > **Review response continued**
> >
> > - *"the result of the LQM with or without improvement seem to be worse than what SSL techniques also relying on pseudo-labeling ..."*
> >
> > We clarify here that our results are not directly comparable with SSL techniques such as Fixmatch, UDA. Reasons:
> >
> > 1. Our synthetic dataset (the NoisyLabelTrain split) is smaller than the training split of the standard CIFAR10 and CIFAR100 datasets. As mentioned in Section 2.1, we use 50% data to train rater models, and generate noisy label synthetic dataset on the other 50% data. So N is around 50% of the training split of the original dataset. We explained this in a footnote in Section 4 in the revision.
> >
> > 2. We would like to emphasize that our technique does not rely on specific data augmentations. Some state-of-the-art SSL methods use augmentations that are specifically designed for certain data types, e.g., Fixmatch uses pairs of strong and weak data augmentations for images. Our LQM technique, however, is general and can be applied to any type of data. We expect that combining these techniques (data augmentation, consistency training in UDA) and LQM will further improve the results, but since the main goal of this section is to demonstrate the use of rater features and how they help improve the *noisy label* techniques, we think our current set of experiments is sufficient.
> >
> > On the other hand, we agree with you that since our LQM technique requires a small subset with clean labels, it has some similarities with SSL, and we added a paragraph discussing this point in Section 4 in the revision.
> >
> > - *"... convex combination of LQM and noisy label is unmotivated and introduces another hyperparameter"*,  *"it remains completely unclear what the LQM model really learns, there is also no discussion... "*
> >
> > The convex combination is used to prevent the training target from getting too far from the original labels. It is particularly helpful for datasets with a large number of classes such as CIFAR100. We discussed this in the revision.
> >
> > The intuition for LQM is to correct the labels in a rater-dependent manner. This means that by learning the patterns from the paired-subset, we can conduct rater-dependent label correction. For example, LQM can potentially learn that raters with a certain feature often mislabel two breeds of dogs, then it can possibly correct these two labels from similar raters for the rest of the data. Below we formally present the details of LQM. We added a paragraph to discuss this in Section 4.
> >
> > **Clarity** section
> >
> > - *"LQM in Figure 8 means LQM+Bootstrap. Why is then LQM+Bootstrap improving so strongly for high label noise? N is unspecified"*
> >
> > In Figure 8, LQM means simply using LQM to correct the labels and then run vanilla training with cross entropy loss function. LQM + Bootstrap is the third item in the legend of Figure 8.
> >
> > LQM methods typically improve more on high label noise setting since more noisy labels can be corrected using LQM.
> >
> > N is around 50% of the size of the original training split for each dataset as mentioned above. It corresponds to the *NoisyLabelTrain* split. We explained in the revision.
> >
> > - *"In the appendix it remains unclear what kind of data augmentation is used ... the individual models is not provided - in particular what do these ones achieve just training on the clean data"*
> >
> > We only use the standard flipping and cropping data augmentation and did not use more sophisticated augmentations. We mentioned this in the appendix in the revision. We provided the accuracy of average over 5 runs and also reported the standard deviation, which we believe is more common and informative than reporting individual models. We reported the test accuracy with clean data in some results (e.g., the legend in Figure 6). In general, training with clean data will lead to higher accuracy than all other techniques. However, since our focus is noisy label techniques, we believe it is more fair to only compare methods that only have access to noisy labels.
> >
> > **Documentation** section
> >
> > - *"... the paper is also a bit off-topic in this track."*
> >
> > Our paper focuses on how to generate realistic noisy label datasets, which can be used to benchmark new and existing techniques. We do plan to publish the data that we generated for other researchers to use (as provided in the URL private to the reviewers). Thus we believe our paper aligns well with this track.
> >
> > **Additional Feedback** section
> >
> > - *"... emphasize more the part in Section 2 ... The LQM method ... missing comparison to self-training/pseudo-labeling SSL techniques."*
> >
> > We propose this benchmark for evaluating algorithms under realistic noise. Since this benchmark provides access to the underlying rater features, we also provide a simple baseline LQM that leverages this feature of our benchmark, unlike some methods such as independent label flipping. In the revision we emphasized more on Section 2 as well. We added the discussion on SSL techniques in the last paragraph of Section 4 (LQM) in the revision.

---

> > > ### Comment · Reviewer_J3SL · 2021-07-19
> > > **Reply to rebuttal**
> > >
> > > Thanks a lot for the detailed answer.
> > >
> > > - Regarding "qualitative similarity to human raters" or "trend of the distribution rather than individual instances":
> > >
> > >   I understand that the purpose of these plots is just to show that the label noise is quite different than random noise. This statement would be perfectly fine but then the authors insist on that the label noise shows qualitatitive similarity to human label noise and in my point of view this claim is too strong and just examined on one dataset. If the authors want to make this claim then they would have to show:
> > > i) this relation on several datasets
> > > ii) they would have to show how the choice of the neural networks models influences this "similarity"
> > >
> > > In the current state this claim is too strong and not sufficiently experimentally validated and thus should not be published like this.
> > >
> > > - "We intentionally removed the y-axis in Figure 2 since the histograms have different scales. The important message here is that for our dataset and CIFAR10-H, the shapes of the distribution histograms are both right-skewed, whereas random label flipping has a different shape (concentrated in the middle)."
> > >
> > > One can normalize a histogram - I don't see why there is a problem of scale. The more important question is why the x-scale is not chosen comparable. This would have been easy to fix in the revision. I doubt that in the same scale the qualitatitive similarity of human and synthetic raters would be so strong (clearly it will be different to random raters).
> > >
> > > - "We clarify here that our results are not directly comparable with SSL techniques such as Fixmatch, UDA. Reasons: "
> > >
> > > It is an important question how good one can get with a small set of high quality labels vs. a larger set of noisy labels. Thus a strong SSL baseline would have been useful -  in particular since SOTA SSL approaches use pseudo-labels which is the approach taken also in this paper. Fixmatch achieves with 250/4000 labels on CIFAR10 an accuracy of 94.9% resp. 95.7% and on CIFAR100 with 2500 labels an accuracy of 71.7%.  As far as I can judge the number of clean labeled data used by the authors for the LQM training is more or on the same order than what Fixmatch uses, so the  SSL results are important to judge how well learning with noisy labels works vs. similar SSL approaches trained using much less but clean labeled data. As Fixmatch outperforms LQM by a large margin, this has to be discussed appropriately.
> > >
> > > I appreciate the effort of the authors but given the open points above and the thoughtful review of reviewer GrzE, in my point of view this paper is not ready yet for publication and thus I keep my score.

---

### Official Review · Reviewer_GrzE · 2021-07-05
**Good direction, but needs more work: no comparison with prior work, potential issues with assumptions, lacks justification that label noise is realistic**

**Rating:** 4
**Confidence:** 5
**Clarity:** The paper is extremely clear and well…

**Strengths:**

The strength of this paper lies in Section 3 and Section 4. Section 3 is the first I've seen to study the effects of both class imbalance and pretraining on learning with labels generated by psuedo-label learning approaches. I thoroughly enjoyed these sections of the paper. Similarly, Section 4 pushes the field of synthetic-annotations by drawing additional ties between multiple synthetic annotators (from rater models) versus multiple human annotators (in terms of history and quality). The results of Section 4 seem useful contributions to dealing with underspecification (D'amour et al. (2020)) and looking at empirical examples of model-based item response theory (versus student/human-based). Both Section 3 and Section 4 provide contributions which seem useful to the broader research community.

**Weaknesses:**

A central claim of this paper is the ability to generate realistic noisy labels, similar to multiple-annotated human generated noisy labels for a given dataset (Section 2). I am concerned that the evidence presented in the paper is insufficient to infallably validate this claim.

A possible immediate issue is that CIFAR-10 is assumed to be clean in this paper (L79 - c.f., "ground truth labels") but CIFAR-10 has been shown to contain label errors (see: https://labelerrors.com/?dataset=CIFAR-10). CIFAR-10 includes errors in the test set which is especially problematic given L89 in the paper.

Specifically, I am concerned that realistic/human-like noise generation claim in the paper (c.f., Figure 1 (b) which shows that the predictions from a large-parameter ResNet can represent a more-expert human rater and a small-parameter MLP's predictions can represent a more-novice human rater) may sometimes be false for CIFAR-10 because CIFAR-10 contains noisy labels. Several prior works suggest this might be the case, for example, (Advani, Saxe, & Sompolinsky (Neural Networks, 2020) and (Belkin, Hsu, Ma, & Mandal (2019)) show that smaller models (resnet-18) **actually outperform** larger models (resnet-50) on datasets with noisy labels because the smaller models regularize against label noise during training. Similarly (Zhang, Bengio, Hardt, Recht, & Vinyals (ICLR, 2017)) showed that larger-parameter models are better at over-fitting to the label noise (and can exactly predict random labels with near-perfect accuracy). Similarly, we've seen evidence that smaller models can outperform larger models if the test set is sufficiently noisy when trained on CIFAR-10 with no noisy labels added (Northcut et al., ICLR (2021)). These works suggest simple models may sometimes perform on par with complex models when the dataset contains noisy labels, which goes against the claims of realistic data because of the expert/novice human rater analogy argument in this paper. I think this needs to be addressed (even briefly) for the paper to be accepted.

Some ways the authors might reconcile these issues are to: (1) use the corrected test set release for CIFAR-10 and also correct errors in the train set; (2) ignore error in CIFAR-10 and justify why that's a reasonable thing to do in terms of issues raised above; (3) possibly use your LQM approach to remove the errors in both the train and test sets.

Figure 3 seemed like a good way to substantiate the claim that the noisy-label generation process is human-like with empirical evidence, but in Figure 3, the off-diagonals of these matrices (which quantify the class-conditional label noise) appear significantly different in distribution in terms of real noise versus synthetic noise. If the paper wants to lean on empirical evidence to corroborate the claim of generating synthetic but realistic noise, that seems fine, but Figure 3 would need to more closely match real versus synthetic across a few diverse settings. Figure 3 is also missing the absolute difference of subfigure (a - human annotators) and subfigure (b - synthetic annotators) which is needed to quantify how close to 'realistic' this noise generation process is. Because no theoretical justification is provided and the main claim is supported empirically, it is also necessary to compare more than just the CIFAR-10 dataset to ensure that the empirical findings in Figure 3 generalize and the method presented can be used to synthetically produce noisy labeled benchmark data for other datasets.

For minimal acceptance the authors must provide empirical and/or theoretical justification to infallibly demonstrate that the predictions from multiple models trained with different parameter settings is a reasonable proxy for the labels assigned by human raters of various levels of expertise.

For a strong acceptance, guarantees about the conditions when the rater model predictions actually produce realistic label noise is needed. (along with a clear description of a realistic noise model)

Because a central claim of the paper (and the title) is that the generated noisy annotations are realistic/human-like, the above issues seem (to me) a major impedance to acceptance.

A (very important) weakness is the paper is the missing a comparison of the authors' noise generation approach vs prior baseline approaches for realistic label noise generation (some of which are easy to compare against via open-source packages, e.g. cleanlab). See "Relation to Prior Work" below.

A precise mathematical description of the noisy label generation process (in terms of assumptions and conditions) is missing. This is necessary since the purpose of creating these datasets is benchmarks and researchers need to know which methods to benchmark, e.g. symmetric noise? asymmetric noise? etc? Various methods address various settings. Which settings can this paper cover and can the authors make it precise?

**Additional Feedback:**

Specific feedback:

L17 - "The first step of research on label noise is to identify the appropriate datasets." -  Remove this claim as it does not strengthen the paper and its not necessarily true. Most people would say the first step is to define a noise model (in your case, instance-dependent). I would recommend starting this paragraph without a claim and instead summarize the main message of the paragraph in the first sentence (e.g., use L22 to start the paragraph, which makes the main (and interesting!) claim)

L30 - " we often have additional features of the raters, such as tenure, historical biases, and expertise level." - Citations to support this claim would help support the contributions that follow.

L64 - TYPO: "We find that the behavior of these techniques on the our synthetic datasets"

L103 - Why is this set of four criteria the right criteria? Is there a principled approach to coming up with these criteria?

L132 - Very glad to see the inclusion of the CIFAR10-H dataset here! Evaluating this dataset against your method is a highlight of this paper and might be greater emphasized. I would consider mentioning this result in the abstract.

L160 - Careful here... these papers assume symmetric uniform class-conditional label noise. They don't show that deep learning is robust to asymmetric class-conditional label noise.

Fig 3 - We need to see the diff between (LEFT) and (MIDDLE). Its too hard to compare by eye. They should be similar enough to support the claim of generating realistic or human-like noisy labels.

L215 - precisely define 'learnability'

L221 - TYPO: change 'by' to 'with' (or 'using' or 'via')


How does this work related to underspecification (D'amour et al. (2020))? Did the authors study how their various rater models tended to agree/disagree on predictions and why their approach is reasonable? Is it possible that one of the rater models is simply poorly configured such that it produces systematic error (e.g., always predicts only a few of the classes)? How do the authors control for this to ensure their generated datasets are useful?


This new datasets and benchmarks track is an important (and much needed) step taken by the NeurIPS committee to encourage a deeper understanding of data-centric methods, datasets, and benchmarks. I think an analysis of how the noise generation process in this paper alters distribution of label errors (the actual data, not just performance in models) would be very appropriate for this track. Some examples of works that focus on the dataset perspective include Shankar et al. (2020), Beyer et al. (2020), Recht et al. (2019), Tsipras et al., (2020), Taori et al. (2021), Liu, Niles-Weed, et al. (2020) ). More time spent on understanding how their generation methodology characterizes the data and an analysis of how dataset characterization from their generation method is linked to benchmarks could be helpful.


**Correctness:**

I mention several issues above. Additionally, there seems to be an error in nomenclature used in the paper regarding prior work. The authors user the term "instance-dependent" to describe the label noise studied. But throughout the paper, the results emphasize asymmetric class-conditional label noise. This occurs in Figure 3, Criteria (b) on L107, etc. Clarification is needed and a mathematical description of the noise process studied would greatly improve the paper. However, one cannot study "instance dependent" label noise without making assumptions to clarify aleatoric (label noise) from epistemic label noise (inherent model noise in predicted probabilities). I agree that the authors' multiple-rater pseudo-label noisy label generation process doesn't inherently make a class-conditional assumption and so I understand why the authors may call it 'instance dependent', but the analysis in the paper is not instance-dependent, so I think the authors should either (1) state the asymmetric class-conditional noise assumption of the paper used for the figures and claims or (2) clarify the assumptions used for instance-dependent noise and precise/mathematically define the label noise process.

**Documentation:**

I looked at the cifar-10 and cifar-100 dataets uploaded by the authors. I may have missed it, but I had trouble finding an index mapping from their weakly labeled data to the original datasets? It would help to make this mapping clear in the colab/ipynb example if it exists. The authors should provide a way to index the examples and their weakly labels back to the original datasets for reproducibility (otherwise we cannot verify the origin of the examples). (if the authors did this and I just missed it -- apologies!)

Also, two small readability things in your jupyter lab / colab demo:
1. add `plt.show()` to the end of your looping through the images and printing them out otherwise it will show the last one.
2. add the class names and print them out to make the demo human readable.

**Relation To Prior Work:**

This aspect of the paper needs significant improvement. Figure 3, Criteria (b) on L107, etc. refer to assymmetric class-conditonal noise processes, but no prior works are mentioned (in fact, the terms 'assymmetric' and 'class-conditional' are never mentioned in the paper). Some related works include: SCE-Loss (Wang et al., 2019), Mixup (Zhang et al., 2018), MentorNet (Jiang et al., 2018), Co-Teaching (Han et al., 2018), etc.

The biggest issue here for the paper is the missing comparison of the authors' noise generation approach vs prior baseline approaches for realistic label noise generation (some of which are easy to compare against via open-source packages, e.g. cleanlab). The authors instead compare LQM versus a vanilla model which avoids this much-needed comparison with prior art. There are several straightforward baselines for generating realistic noisy labeled datasets (the problem this paper solves) which would strengthen the paper by explicating the contributions of LQM compared to prior art. Three suggestions for baselines:
(1) Perhaps the simplest (and easy to implement) class-conditional is to compute the confusion matrix of predictions versus given labels for a dataset and use that as an approximation of the joint distribution of noisy and true labels. Several prior works consider this baseline (e.g. Hendrycks et al. (NeurIPS, 2020), Sukhbaatar et al. (ICLR, 2015), Chen et al., (ICML, 2019)).
(2) The confident learning (Northcutt, Jiang, & Chuang (2021)) approaches provide a standard benchmark comparison (see cleanlab python package). They model the joint distribution of noisy and true labels for a given dataset which can increase/decrease the amount of label noise proportionally to the distribution of real-world label noise in the dataset (by generating the noisy labels based on this noise-increased/noise-decreased joint distribution of noisy and true labels). Confident learning approaches have some theoretical guarantees for exact estimation of the distribution of label noise.
(3) A comparison with Lu Jiang, Huang, Liu, & Yang (ICML, 2020) realistic synthetic noise generation approaches. These folks are also at Google, so it might make collaboration easier for the authors.

The LQM approach used in this paper is very similar to Item Response Theory, which is not discussed. Item Response Theory (IRT) is a latent-trait model useful for estimating learner-proficiency and example hardness/difficulty. IRT is often overlooked in machine learning papers (probably because it originally was developed in psychometrics and measurement theory); I think the authors will be particularly interested in the principled nature of IRT for using multiple raters/learners to estimate the hardness/difficulty of an instance-dependent labeled-example similar to this paper which uses multiple rater models to estimate/generate noisy labels.

Why is there no mention of Dawid & Skene's work on multiple annotators? The goal of Section 4 is to show that the authors' LQM model leverages synthetic-annotator information. Dawid & Skene were the first to do this for human-annotator information.

That (1) Dawid & Skene's work is not mentioned, (2) the nomenclature on label noise processes has issues, and (3) prior work on synthetic noise generation is not compared with; may suggest that this work needs to mature further before it is ready for publication.

**Summary And Contributions:**

This work focuses on creating realistic benchmark datasets with noisy labels. Noisy labeled datasets are generated by combining predictions from multiple models (trained with different hyper-parameter settings) and using these predictions as noisy labels. The paper claims this process produces realistic noise labels similar to the noisy labels multiple human annotators might generate. Evidence from many prior publications suggests that **this claim may sometimes be false** for datasets with noisy labels (see *Weaknesses section* below).  Further, no comparison of their methods with prior works for finding noisy labels in datasets or weak supervision (two different tasks), is included. Without some kind of comparison, it is difficult to quantify how the contributions of this work sit in the context of prior work.

Also, the background/prior work covered is lacking (perhaps severely lacking). For example, one might summarize the primary contributions of Section 1 and Section 2 as simple saving the labels of ensembles [3], yet no coverage of ensembles is mentioned. This is also related to distillation (training a single model to distill the knowledge of ensembles) [1], [2], but distillation is similarly not mentioned in this context. There was also no mention of Dawid & Skene's seminal work on multiple annotations [4]. Further, the coverage of learning with noisy labels and data cleaning approaches is lacking (specific details provided in sections below).

On the positive side, I found the contributions in Section 3 and Section 4 enlightening, namely that noisy labels are more detrimental under class imbalanced settings, when pretraining is not used, and on tasks that are easier to learn with clean labels (Section 3) and their LQM model that strengthens the ties between prior work with human annotations and their work on model-synthetic annotations (Section 4). Sections 3 and 4 were the highlights of the paper from my perspective.


[1] Geoffrey Hinton, Oriol Vinyals, and Jeff Dean. 2015. Distilling the knowledge in a neural network. arXiv preprint arXiv:1503.02531 (2015).

[2] Cristian Buciluˇa, Rich Caruana, and Alexandru Niculescu-Mizil. 2006. Model compression. In Proceedings ofthe 12th ACM SIGKDD international conference on Knowledge discovery and data mining. 535–541.

[3] Thomas G Dietterich. 2000. Ensemble methods in machine learning. In International workshop on multiple classifier systems. Springer, 1–15.

[4] A. P. Dawid and A. M. Skene. Maximum Likelihood Estimation of Observer Error-Rates Using the EM Algorithm. Applied Statistics, 28(1):20–28, 1979

---

> ### Author Response · Authors · 2021-07-13
> **Review response -- we believe all the issues have been clarified in the response and/or addressed in the revision**
>
> Thanks for your time and detailed comments. We believe all the issues that you raised have been clarified in this response and/or addressed in the revision in our local file. The revised version will be uploaded once we finish replying to your review. Here, we provide a point-by-point response to your comments. We are also happy to discuss more during the discussion phase, and we hope you can increase your score after reading our response and revision.
>
> **Summary And Contributions** section
>
> -*"the background/prior work covered is lacking ... ensembles ... distillation ... Dawid & Skene ...  data cleaning approaches ..."*
>
> We added discussions on ensemble, knowledge distillation, Dawid and Skene's EM algorithm and a few follow-up works, the confident learning (CL) method (a data cleaning appraoch) in Section 5 in our revision.
>
> **Weakness** section
>
> -*"CIFAR-10 includes errors in the test set"*, *"... concerned ... Figure 1(b) ... smaller models ... actually outperform larger models"*, *"Some ways the authors might reconcile ... ignore error in CIFAR-10 and justify why that's a reasonable thing"*
>
> We appreciate the reviewer for pointing out the cleanlab package and the link to the labelerrors.com website. We notice that this package corresponds to the paper by Northcut et al., 2021, which was first uploaded to arXiv on March 26, 2021. This is a relatively new paper and we were not aware of it when we were working on the draft of our paper.
>
> We agree that it is possible that many public datasets have incorrect labels. Therefore, we avoided the term “ground truth” in the revision. In the Section 2.2 (Dataset generation) in our revision, we discussed the Northcut et al., 2021 paper and acknowledged that there can be noisy labels in public datasets.
>
> However, **the amount of noise in these datasets is much smaller than the typical amount of noise that the noisy label research community considers**.  Northcut et al., 2021 mentioned that they estimate an average of 3.4% errors across the 10 datasets. In particular, we noticed that according to the labelerrors.com website, there are only 22 examples in the CIFAR10 test set (10,000 samples) that have incorrect labels. In our paper, the three synthetic datasets for CIFAR10 have 11%, 19%, and 48% noisy labels, significantly higher than the amount of label noise in the original dataset. Similar amount of label noise has been considered in prior work on noisy labels (see our discussion in Sec 2.2 Dataset generation). Therefore, we believe that it is fair to consider labels in public datasets as “clean labels”, and we do not expect the label noise in the original datasets to change our conclusions.
>
> Regarding Figure 1(b), we believe that there might be a **misunderstanding**. Throughout the paper, **we did not use the assumption that models with larger numbers of parameters such as ResNet are more human-like**. All we need is a pool of rater models that make possibly different predictions on the NoisyLabelTrain and NoisyLabelValid splits. This can be achieved by choosing different architectures and/or hyperparameters. The only purpose of Figure 1(b) is to make an analogy between the pool of rater models and a pool of human raters. Since this figure caused some confusion, we removed it in the revised version.
>
> We appreciate the reference to a few papers mentioning that smaller models can outperform larger models in the presence of label noise. We mentioned them in the first paragraph of Section 3 (impact of label noise).

---

> > ### Author Response · Authors · 2021-07-13
> > **Review response continued**
> >
> > -*"Figure 3... missing the absolute difference of subfigure (a - human annotators) and subfigure (b - synthetic annotators)..."*
> >
> > We would like to emphasize that the goal of our simulation framework is to generate synthetic noisy labels that are as realistic as possible, rather than making the noisy labels or class confusion matrix exactly the same as certain datasets collected from human raters. **The main purpose of Figure 3 is to show that the label noise in our synthetic dataset is class-conditional.** This also holds true for human labeled datasets such as CIFAR10-H, but does not hold for symmetric random noise. In the revision, we clearly stated that the purpose for Figure 3 is to verify the class-conditional property.
> >
> > We don’t think it is necessary to make the confusion matrix close to that of CIFAR10-H in absolute value for each entry. First, our framework is controllable. Users can generate different amounts of label noise for their research purposes. In fact, the rater error rate of CIFAR10-H is around 5%, much smaller than the rate error rate that most noisy label literature considers. Second, making the predictions of neural networks similar to humans is an interesting problem but we believe that it is still widely open (consider adversarial examples as an extreme case), and is beyond the scope of our paper. Based on these reasons, **we removed the class confusion matrix of CIFAR10-H and the symmetric noise case** in our revision and only left our synthetic dataset, and emphasized that our goal for this figure is to show that the label noise is class-conditional. We also added a figure for the class confusion matrix of a CIFAR100 synthetic data in the revision.
> >
> > -*"For minimal acceptance ... for a strong acceptance, guarantees about the conditions when the rater model predictions actually produce realistic label noise is needed. (along with a clear description of a realistic noise model)"*
> >
> > In the revision, **we added a new section (Sec 2.1 Formulation)**, which theoretically defined three approaches to generating noisy labels:
> > 1. Independent random flipping (or the symmetric label noise), i.e., with probability delta, the label is flipped to an incorrect one which is uniformly chosen;
> > 2. Class-conditional (asymmetric) case, where the noisy label still only depends on the clean label, but is flipped according to a transition matrix
> > 3. The most general instance-dependent case, where the distribution of noisy labels depends on the input feature and rater as well. Clearly our framework satisfies the instance-dependency criterion, whereas many prior methods do not.
> >
> > -*"A precise mathematical description ... Which settings can this paper cover and can the authors make it precise?"*
> >
> > As mentioned above, we added Section 2.1 (Formulation) which clearly discussed the mathematical description of the noisy label generation process and the desired property of realistic noise generation.
> >
> > **Correctness** section
> >
> > -*"... the term 'instance-dependent' to describe the label noise studied ... the authors should either (1) state the asymmetric class-conditional noise assumption ... or (2) clarify the assumptions used for instance-dependent ..."*
> >
> > As mentioned above, we added Section 2.1 which clearly distinguishes between “class-conditional (asymmetric)” noise and “instance-dependent” noise. We updated Figure 3 and acknowledge that this figure only verifies the class-conditional property.
> > In our original submission, in the second paragraph of the introduction, we discussed several prior works that generate noisy labels according to a probability transition matrix, which corresponds to your class-conditional or asymmetric case. We did mention that the label noise in these works does not depend on the input feature. We believe in the revision, we made this point much clearer.
> >
> > **Relation To Prior Work** section
> >
> > -*"assymmetric class-conditonal noise processes ... some related works include ..."*
> >
> > In the revision, in Section 2.1, we have a thorough discussion on class-conditional noise vs our noise generation and discussed these related works.
> >
> > -*"prior baseline appraoches ... Three suggestions for baselines: (1) Perhaps the simplest ... class-conditional "*
> >
> > We already have a discussion on this in the second paragraph of the introduction. In the revision, we clearly discussed this point in Section 2.1 Formulation, and discussed these related work.
> >
> > -*"(2) confident learning..."*
> >
> > We added a discussion on confident learning in Section 5.
> >
> > -*"(3) A comparison with Lu Jiang, Huang, Liu, & Yang (ICML, 2020) ..."*
> >
> > We already compared this work in Section 5 of our original submission.
> >
> > -*"... Item Response Theory ... Dawid & Skene's work on multiple annotators ..."*
> >
> > We added discussion on IRT and Dawid & Skene's work (and a few follow-ups) in Section 5 of the revision.

---

> > > ### Author Response · Authors · 2021-07-13
> > > **Review response continued**
> > >
> > > **Documentation** section
> > >
> > > -*"cifar-10 and cifar-100 dataets ... mapping from their weakly labeled data to the original datasets"*, *"...readability in demo..."*
> > >
> > > Unfortunately at this point it is hard to get the index mapping to the original datasets for CIFAR10 and CIFAR100. Both the training and validation splits are subsets of the training splits of CIFAR10/100. This won't affect the use of our datasets for future research since we already provided the images and noisy/clean labels. For PCam and Cats vs Dogs, we do have the index mapping to the original datasets. For the demo, our open sourcing process is still ongoing and we will incorporate your suggestions when we make the data and demo publicly available.
> > >
> > > **Additional Feedback** section
> > >
> > > -*"Specific feedback: L17 ... L221 ..."*
> > >
> > > We have addressed all of these issues in our revision.
> > >
> > > -*"... underspecification (D'amour et al. (2020)) ... various rater models tended to agree/disagree on predictions... rater models ... produces systematic error ..."*
> > >
> > > Designing instance-dependent noisy label generation methods can be considered as reducing underspecification in the noisy label simulation framework, since we only specify the noisy labels based on the input features. We discussed this in Section 5 of the revision. As mentioned in the paper, we can use the CleanLabelValid split to check the quality of the rater models. In our experiments, we can identify scenarios where rater models tend to agree more (high k-alpha) or disagree more (low k-alpha) on predictions.
> > >
> > > -*"...analysis of how the noise generation process in this paper alters distribution of label errors..."*
> > >
> > > We have a new section (Sec 2.1 Formulation) to discuss how the noise generation process differs from prior works, and how instance-dependent label error generation should work.

---

> > > > ### Comment · Reviewer_GrzE · 2021-07-21
> > > > **Reply to Rebuttal**
> > > >
> > > > Thank you for the thorough response. I'm glad to see many of these edits, the increased use of language in the field and the inclusion of the additional related work.
> > > >
> > > > Overall,
> > > >
> > > > * One of the main contributions of the paper is saving the predictions of various models (with various hyper-parameter settings) and releasing it as a dataset. While its nice to have a standard version with the same labels, this is a procedure that most of us (who benchmark datasets with models) do and I'm not sure this is the right fit in terms of novelty and appropriateness for a paper acceptance in this dataset track at NeurIPS... plus the assertion that the label noise is **realistic** is a huge claim and the paper still fails to provide infallible/sufficient evidence to backup this big claim.
> > > >
> > > > * The authors mentioned that because public datasets like CIFAR-10 have very few label errors, the label errors in these datasets do not matter because in this work this paper focuses on higher proportions of label errors. This misses the point in two different ways. First, this paper trains models on the **clean dataset CIFAR**, prior to generating the noisy labels (but CIFAR-10 isn't clean). The issue with the label errors in CIFAR-10 has to do with this process (not the resulting predictions dataset). Second, the issues with benchmark rankings becoming destabilized do not occur for low noise, they actually occur for an **increase** noise (c.f., Figure 4 and Figure 5, Northcutt, Athalye, & Mueller). The authors make a good point that this is recent work and they did not have time to take this into consideration, which is reasonable.
> > > >
> > > > * I think the authors did well to significantly increase their coverage of prior work and relation to prior work based on the feedback in this review process. I think this has strengthened the paper and I hope they feel the same. However, it remains that the paper still lack quantitative benchmark comparison with prior art. They show their method combined with MentorMix and Co-Teaching (e.g. Fig 8), which is a start, but it would strengthen the paper to have a **direct comparison**, even in some controlled capacity, to argue (more than just qualitatively) how their work sits in prior art.
> > > >
> > > > For these reasons, I maintain my score as is. However, I think the revised paper is already an improvement (in terms of contextualization in the field) and the authors should consider submitted to Round 2 after revising based on the above suggestions and the suggestions from Reviewer J3SL. I think with additional maturity, this work could be a viable contribution.

---

### Official Review · Reviewer_DP6J · 2021-07-06

**Rating:** 7
**Confidence:** 3
**Correctness:** Experiments are very careful and sens…
**Clarity:** Yes.

**Strengths:**

The Label Quality Model seems like a useful general purpose tool.

Interesting connection between the study of label noise and semi-supervised learning, which hadn't occurred to me before.

I didn't know label noise was interesting to study (my thought was "label noise sucks, but what can you do about it?")---this paper has convinced me that it is worthwhile to assume that your dataset has label noise and to try applying LQM and seeing if results improve.

Nice paper, well written.

Code open-sourced.

**Weaknesses:**

"Training machine learning models typically requires a large amount of labeled data" --> this is a pretty broad/very often not correct statement

The LQM is presented in a generic probabilistic way, but I think there are some important cases where it might not make sense, like variable-length labels in speech recognition or machine translation, where it is not possible to "interpolate" between two possible labels. Could be interesting to extend the model to this case.

Setting the interpolation hyperparameter for the LQM uses the validation set labels---but how do we know that these don't have noise too? Would it be worthwhile or possible to study the impact of label noise on the validation set as well?

**Additional Feedback:**

n/a

EDIT: I'm actually not sure this submission is in scope for the Datasets and Benchmarks track? "* Papers that mainly introduce a new method and don't really address any of the above are a better fit for the main NeurIPS track." Still, I'll leave my score as-is and let the ACs decide.

**Documentation:**

Yes.

**Ethics:**

No.

**Relation To Prior Work:**

The paper carefully describes related work and how theirs improves upon it, though I don't know enough about this topic to say whether they accurately report prior work.

**Summary And Contributions:**

The authors propose a more realistic model of label noise, in which pseudo-labels are generated by models trained on a subset of data. Typically label noise is modelled by simply randomly flipping the label, which does not actually match real-world label noise (e.g., semantically similar classes are more likely to be confused). They also suggest a "Label Quality Model" that conditions on the quality of a labeler and can be used to generate more robust labels, and find that it helps over a variety of settings. (Note: I am an "emergency reviewer" for this paper, and this is not my area of expertise.)

---

> ### Author Response · Authors · 2021-07-08
> **Review response**
>
> Thank you so much for your review and positive feedback.
>
> Regarding the first sentence, we have changed it to "In many applications, training machine learning models requires labeled data." in our local file.
>
> The extension to variable-length labels is very interesting and will be our future work.
>
> The validation set also has noise. We use the NoisyLabelValid split (see Figure 1(a)) for hyperparameter tuning. It is definitely true that the noise in the validation set can affect the hyperparameters that we select. This is also a very interesting future direction, but it is currently beyond the scope of our paper. We added a footnote which mentions that the noise in the validation set can affect the hyperparameter selection in our local file.
>
> We will upload the revised version once we finish addressing all the comments. We are happy to discuss more during the discussion phase.

---

> > ### Comment · Reviewer_DP6J · 2021-07-13
> > **Response**
> >
> > Thanks for your response and paper update! I'll keep my score at 7.

---

### Author Response · Authors · 2021-07-13
**Revision has been uploaded. We believe all the issues that reviewers mentioned have been clarified in the response and/or addressed in the revision.**

Thanks again to all the reviewers for their time and comments. We have carefully responded to all the comments from every reviewer, and a revised version has just been uploaded.

We observe that all the reviewers agree that our proposed method is a solid contribution to the noisy label research community, and our paper is well-written. In fact, we did not see any concerns about the design of our noisy label generation approach and our experiments. The issues that the reviewers raised, in particular, reviewers GrzE and J3SL, are mainly clarification questions, (potentially) misunderstanding, and additional related works. We believe we have fully addressed these issues in our revision.

A few highlights in the new version:
1. A new section (Sec 2.1) with a rigorous description of the mathematical formulation of the noisy label generation process. We clearly distinguished among "independent random flipping", "class-conditional", and "instance-dependent". We believe this section addressed one issue raised by reviewer GrzE.
2. A revised version of dataset evaluation (Sec 2.3). We believe the revised version is more precise and clear about the evaluation results, which we believe resolves the concerns from reviewers GrzE and J3SL.
3. An extended version of Section 4 (LQM), which addressed some issues mentioned by reviewer J3SL.
4. An extended version of Section 5 on additional related work. We believe we have discussed all the related works mentioned by reviewer GrzE.

We hope the reviewers, in particular reviewers GrzE and J3SL, can read our response and revised version and hopefully increase their scores. We notice that there is still some time before the discussion phase ends, so we are happy to discuss more with the reviewers, if there is anything that we have not addressed or if there is any other clarification questions.

---

> ### Author Response · Authors · 2021-07-13
> **A minor update with a figure for class confusion matrix for CIFAR100 in Section 2.3**
>
> A minor update to the revision has been uploaded as the title suggests.

---

### Decision · Program_Chairs · 2021-07-26

**Decision:**

Reject

**Comment:**

This paper proposes a simulation framework for learning with instance-dependent label noise, which they do via a pseudo-labeling paradigm, evaluating on the CIFAR10-H dataset. The authors also propose a modeling step based on "annotator" features.  While there were several strong points to this paper, the reviewers were not convinced that the central claim of the label noise being accurate was sufficiently proven, or all the label noise/modeling techniques appropriately described.